# Development and validation of the Michigan Chronic Disease Simulation Model (MICROSIM)

James F. Burke[1]*, Luciana L. Copeland[2], Jeremy B. Sussman[3,4,5], Rodney A. Hayward[3,4,5], Alden L. Gross[6], Emily M. Briceño[3,7], Rachael Whitney[3], Bruno J. Giordani[8], Mitchell S. V. Elkind[9,10], Jennifer J. Manly[9,11], Rebecca F. Gottesman[12], Darrell J. Gaskin[13], Stephen Sidney[14], Kristine Yaffe[15], Ralph L. Sacco[16], Susan R. Heckbert[17], Timothy M. Hughes[18], Andrzej T. Galecki[3,19], Deborah A. Levine[3,4]

1 Department of Neurology, The Ohio State University Wexner Medical Center, Columbus, OH, United States of America, 2 Independent Software Developer, Ann Arbor, MI, United States of America, 3 Department of Internal Medicine and Cognitive Health Services Research Program, University of Michigan (U-M), Ann Arbor, MI, United States of America, 4 Institute for Healthcare Policy and Innovation, U-M, Ann Arbor, MI, United States of America, 5 Ann Arbor Veteran's Affairs Hospital, Center for Clinical Management and Research, Ann Arbor, MI, United States of America, 6 Department of Epidemiology, Johns Hopkins Bloomberg School Public Health, Baltimore, MD, United States of America, 7 Department of Physical Medicine and Rehabilitation, U-M, Ann Arbor, MI, United States of America, 8 Department of Psychiatry & Michigan Alzheimer's Disease Center, U-M, Ann Arbor, MI, United States of America, 9 Department of Neurology, Vagelos College of Physicians and Surgeons, Columbia University, New York, NY, United States of America, 10 Department of Epidemiology, Mailman School of Public Health, Columbia University, New York, NY, United States of America, 11 Taub Institute for Research on Alzheimer's Disease and the Aging Brain, Columbia University, New York, NY, United States of America, 12 Stroke Branch, National Institute of Neurological Disorders and Stroke (NINDS), Bethesda, MD, United States of America, 13 Department of Health Policy and Management, Johns Hopkins Bloomberg School of Public Health, Baltimore, MD, United States of America, 14 Kaiser Permanente Northern California Division of Research, Oakland, CA, United States of America, 15 Departments of Psychiatry, Neurology and Epidemiology, University of California, San Francisco, San Francisco, CA, United States of America, 16 Department of Neurology, Miller School of Medicine, University of Miami, Miami, FL, United States of America, 17 Department of Epidemiology, University of Washington, Seattle, WA, United States of America, 18 Department of Internal Medicine, Wake Forest School of Medicine, Winston-Salem, NC, United States of America, 19 Department of Biostatistics, U-M, Ann Arbor, MI, United States of America

* James.Burke@osumc.edu

**Data Availability Statement:** All code for MICROSIM is available at: https://github.com/jburke5/microsim All code and results for the

## Abstract

Strategies to prevent or delay Alzheimer's disease and related dementias (AD/ADRD) are urgently needed, and blood pressure (BP) management is a promising strategy. Yet the effects of different BP control strategies across the life course on AD/ADRD are unknown. Randomized trials may be infeasible due to prolonged follow-up and large sample sizes. Simulation analysis is a practical approach to estimating these effects using the best available existing data. However, existing simulation frameworks cannot estimate the effects of BP control on both dementia and cardiovascular disease. This manuscript describes the design principles, implementation details, and population-level validation of a novel population-health microsimulation framework, the MIchigan ChROnic Disease SIMulation (MICROSIM), for The Effect of Lower Blood Pressure over the Life Course on Late-life Cognition in Blacks, Hispanics, and Whites (BP-COG) study of the effect of BP levels over the life course on dementia and cardiovascular disease. MICROSIM is an agent-based Monte

validation analyses are available at: https://github.com/jburke5/microsimNHANES.

**Funding:** This research project is supported by a grant R01 NS102715 from the National Institute of Neurological Disorders and Stroke (NINDS), National Institutes of Health (NIH), Department of Health and Human Service (DHHS). The NINDS was not involved in the design and conduct of the Study; collection, management, analysis, and interpretation of the data; preparation, review, or approval of the manuscript; and decision to submit the manuscript for publication. The content is solely the responsibility of the authors and does not necessarily represent the official views of the National Institute of Neurological Disorders and Stroke or the National Institutes of Health. Additional support was provided by the National Institute of Aging (NIA) grant R01 AG051827 and 1 RF1 AG068410-01 (DL), NIA Claude Pepper Center grant P30 AG024824 (AG), NIA grants K01 AG050699 (AG), and NIA Michigan Alzheimer's Disease Research Center grant P30 AG053760 (BG). Cohort Funding/Support: The Atherosclerosis Risk in Communities (ARIC) study has been funded in whole or in part with federal funds from the National Heart, Lung, and Blood Institute (NHLBI), NIH, DHHS, under Contract nos. (HHSN268201700001I, HHSN268201700002I, HHSN268201700003I, HHSN268201700004I, HHSN268201700005I,). Neurocognitive data are collected by U01 2U01HL096812, 2U01HL096814, 2U01HL096899, 2U01HL096902, 2U01HL096917, and R01 AG040282 from the NIH (NHLBI, NINDS, NIA, and National Institute on Deafness and Other Communication Disorders). Further information about ARIC's funding can be found at https://sites.cscc.unc.edu/aric/desc. The authors thank the staff and participants of the ARIC study for their important contributions. The Coronary Artery Risk Development in Young Adults Study (CARDIA) is conducted and supported by the NHLBI in collaboration with the University of Alabama at Birmingham (HHSN268201800005I & HHSN268201800007I), Northwestern University (HHSN268201800003I), University of Minnesota (HHSN268201800006I), and Kaiser Foundation Research Institute (HHSN268201800004I). A neurocognitive ancillary study is funded by the NIH (NIA, R01 AG063887). A full list of CARDIA collaborators and other cohort study information can be found at https://www.cardia.dopm.uab.edu/. This manuscript has been reviewed by CARDIA for the scientific content. The Cardiovascular Health Study (CHS) was supported by contracts HHSN268201200036C, HHSN268200800007C, HHSN268201800001C, N01HC55222, 379 N01HC85079, N01HC85080, N01HC85081,

Carlo simulation designed using computer programming best practices. MICROSIM estimates annual vascular risk factor levels and transition probabilities in all-cause dementia, stroke, myocardial infarction, and mortality in a nationally representative sample of US adults 18+ using the National Health and Nutrition Examination Survey (NHANES). MICROSIM models changes in risk factors over time, cognition and dementia using changes from a pooled dataset of individual participant data from 6 US prospective cardiovascular cohort studies. Cardiovascular risks were estimated using a widely used risk model and BP treatment effects were derived from meta-analyses of randomized trials. MICROSIM is an extensible, open-source framework designed to estimate the population-level impact of different BP management strategies and reproduces US population-level estimates of BP and other vascular risk factors levels, their change over time, and incident all-cause dementia, stroke, myocardial infarction, and mortality.

## Introduction

Alzheimer's disease and related dementias (AD/ADRD) are major causes of death and disability in older individuals. Preventing or delaying AD/ADRD can lead to better survival, less disability, less nursing home use, lower health care costs, and better quality of life. Blood pressure (BP) management is a promising strategy for AD/ADRD prevention and the key element in the primary and secondary prevention of atherosclerotic cardiovascular disease (ASCVD), namely stroke and acute myocardial infarction. Little is known about the effect of BP treatment on the combined outcomes of ASCVD and AD/ADRD at the population level.

Accumulating evidence over the past 20 years has led to important new clinical guidelines for more aggressive treatment of modifiable vascular risk factors. Most policy assessments and simulation models informing these new guidelines mainly or solely consider BP's impact on ASCVD, but not AD/ADRD [1–4]. Yet BP is a strong risk factor for AD/ADRD, and consideration of BP's effect on AD/ADRD and other disease states may influence clinicians' and policymakers' definitions of "optimal" BP treatment. For example, while lowering BP to optimal levels (<120/80 mm Hg) reduces ASCVD events, mild cognitive impairment (MCI), and the combination of MCI and AD/ADRD in individuals with high ASCVD risk [5, 6], this group in the US is relatively small. It is unclear whether lowering BP to optimal levels also reduces CVD and AD/ADRD in the larger group of adults at lower ASCVD risk (e.g., Black individuals age 55 with systolic BP 130–139 mmHg). If higher BP treatment intensity early in the life course has a large effect on late-life cognition and AD/ADRD, then the optimal timing, treatment threshold, and intensity of BP treatment initiation should shift to earlier ages and more intense treatment. Similarly, estimating the independent effects of BP treatment intensity on ASCVD vs. AD/ADRD may enable a more accurate characterization of the impact of different BP treatment policies on quality of life, cost-effectiveness, and societal benefit informing optimal policy.

While large-scale randomized controlled trials (RCTs) are the least biased approach to estimating the effect of BP treatment on ADRD, such trials may be infeasible. Simulation analyses leveraging evidence from published risk models and meta-analyses of RCTs and the best available observational data may be the most effective and practical approach to estimating these effects currently. While microsimulation models of either ASCVD or dementia exist, microsimulation models of both ASCVD and dementia are lacking. This manuscript describes the

N01HC85082, N01HC85083, N01HC85086, 75N92021D00006, and grants U01HL080295 and U01HL130114 from the NHLBI, with additional contribution from the NINDS. Additional support was provided by R01AG023629, R01AG15928, and R01AG20098 from the NIA. A full list of principal CHS investigators and institutions can be found at https://chs-nhlbi.org/CHSOverview. The content is solely the responsibility of the authors and does not necessarily represent the official views of the National Institutes of Health. The Framingham Heart Study is a project of the NHLBI of the NIH and Boston University School of Medicine. This project has been funded in whole or in part with federal funds from the NHLBI, NIH, DHHS, under contract No. HHSN268201500001I. The Northern Manhattan Stroke (NOMAS) study has been funded at least in part with federal funds from the NIH, NINDS by R01 NS29993. The Multi-Ethnic Study of Atherosclerosis (MESA) was supported by contracts 75N92020D00001, HHSN268201500003I, N01-HC-95159, 75N92020D00005, N01-HC-95160, 75N92020D00002, N01-HC-95161, 75N92020D00003, N01-HC-95162, 75N92020D00006, N01-HC-95163, 75N92020D00004, N01-HC-95164, 75N92020D00007, N01-HC-95165, N01-HC-95166, N01-HC-95167, N01-HC-95168, and N01-HC-95169 from the NHLBI, and by grants UL1-TR-000040, UL1-TR-001079, and UL1-TR-001420 from the National Center for Advancing Translational Sciences (NCATS). Cognitive testing at Exam 6 in MESA has been funded by grants R01HL127659 from the NHLBI and NIA R01AG054069 from the NIH. The authors thank the other investigators, the staff, and the participants of the MESA study for their valuable contributions. A full list of participating MESA investigators and institutions can be found at http://www.mesa-nhlbi.org. The funders had no role in study design, data collection and analysis, decision to publish, or preparation of the manuscript.

**Competing interests:** The authors have declared that no competing interests exist.

design principles, implementation details, and population-level validation of a novel population health microsimulation framework, the MIchigan ChROnic Disease SIMulation (MICROSIM), for The Effect of Lower Blood Pressure over the Life Course on Late-life Cognition in Blacks, Hispanics, and Whites (BP-COG) Study of the effect of BP levels over the life course on cognitive decline and dementia. MICROSIM was designed by applying computer programming best practices to create a novel simulation model of ASCVD and AD/ADRD. The initial research purpose of this extensible, open-source framework is to explore a series of questions related to the impact of different BP management strategies on late-life cognition and all-cause dementia, as well as the effects on race differences in all-cause dementia incidence. The ultimate research purpose, though, is to develop an extensible framework that can be applied to a wide variety of chronic disease management questions, as different clinical conditions are added.

## Methods

### Overview of the research purpose and specific motivating study questions

The development of MICROSIM broadly aligns with best practices in systems science research [7]. The primary research purpose of MICROSIM is to use retrospective data to inform the value of prospective population-level BP control strategies on cardiovascular events, dementia and racial disparities in both. While existing population-based simulations address elements of these questions, we are unaware of any simulation that simultaneously addresses them all and has the potential to account for trade-offs between ASCVD and dementia.

Several specific research questions motivated the design of MICROSIM. First, better BP control leads to less cognitive decline and lower dementia risk at a given age but also longer life expectancy due to avoidance of cardiovascular events. Yet, as age is the dominant risk factor for dementia and non-cardiovascular mortality, how do these effects counter-balance? Does BP treatment reduce dementia risk across the life course or mostly delay it? Second, does the substantial relative effect of BP treatment on dementia translate into a meaningful absolute effect of years of life lived without dementia? Third, how do these dynamics vary by race? Black people have a higher incidence of dementia than white people, but also a higher incidence of stroke. Does more intense BP control lead to worsening of race disparities in dementia due to prolonged survival via stroke-prevention? In addition to these population-level questions, we also sought to characterize individual-level variation in these outcomes. For example, how does benefit vary across different baseline risks for cardiovascular disease and dementia and different life courses.

While MICROSIM's initial design addresses these specific questions, we designed it to maximize transparency, optimize code accuracy, and enable extensibility to address various chronic disease management questions.

### Rationale for using a simulation approach

RCTs are the optimal study design to determine the efficacy of different BP treatment algorithms (e.g., treatment intensity, the timing of initiation) and patient selection strategies on combined cognitive and cardiovascular outcomes. However, it is challenging to design RCTs for these questions. Ideally, we would want evidence from large, long-term RCTs to address the effect of BP and BP treatment across the life course on late-life cognition and dementia, but such trials would be at best cumbersome and costly and, at worst infeasible. Such trials would require enormous sample sizes, rigorous case capture, and robust intervention fidelity—all maintained across decades. Further, conducting such a trial is challenging, and meaningful results would likely be decades off. During that time, the treatment paradigm may have

evolved such that the results are no longer meaningful. Even if these problems could be addressed, these studies would not be able to inform patient care in the near term. Shorter-term trials are more feasible but cannot directly determine how short-term treatment decisions influence long-term outcomes since one would have to make assumptions about the importance of short-term surrogate measures and/or make extrapolations from higher to lower risk populations.

Simulation analyses, by capturing the dynamic relationships between key individual-level trade-offs, may be able to provide guidance when clinical trials are infeasible and interim guidance for important clinical questions while waiting for the results of better, long-term trials. Simulations can combine the strengths of the best available data sources. First, by using the best available longitudinal observational data, they can account for complex natural histories and competing risks. Second, by incorporating the best inferences regarding treatment effectiveness (trial-based when available and the best observational estimates [8] when trials are not available), they can credibly account for a range of treatment effects. By leveraging these strengths in a carefully specified framework, simulations may be the best tools to estimate long-term treatment effects, particularly in low-risk populations. Simulations can also serve as "policy laboratories" by exploring the societal consequences of clinical and policy interventions and inform the likely relative yields of such policies.

Yet, simulations suffer multiple limitations. Simulations' intrinsic complexity often makes them seem like "black box science," frequently leading to a lack of transparency and concerns about the accuracy of their results. Moreover, most current models are built individually and fail to build upon one another, limiting progress and potentially amplifying concerns regarding accuracy. Creating the framework for an effective simulation can be complicated, time-intensive, and require the careful evaluation of many individual decisions. Careful consideration of model inputs is critical for the agent-based simulation design described here.

The premise for our simultaneous individual and population-level simulation design is that a single simulation codebase applicable to multiple scientific questions is more efficient and likely to produce more reliable results than multiple simulations for separate questions. A simultaneous individual and population-level simulation design has three specific advantages. First, it enables a relatively straightforward evaluation of both individual- and population-level effects and describes them at granular levels. For example, if one is interested in the effect of BP treatment on population outcomes, a Markov-based simulation would be well-suited to that task using distributions of population-level parameters as inputs. If individual-level variation in outcomes were of interest (e.g., the effect of BP treatment, conditioned on baseline BP levels), it would also be feasible to modify that model to track outcomes across baseline levels of BP. However, extending that model to other individual-level effects would take considerable effort (e.g., the effect of BP treatment conditioned on age, baseline lipid levels, glycemic levels, other treatments, etc.). Our approach estimates a wide variety of individual-level and population-level effects without altering the core simulation codebase by simultaneously characterizing and maintaining approximate correlation structures across various risk factors. Second, the framework's codebase is reusable and easily modifiable to address a broad range of questions. In addition to quantifying individual- and population-level effects of BP treatment on race differences in late-life cognition and dementia, a secondary goal of our project is to explore possible clinical trial designs to assess the effect of BP treatment on late-life cognition and dementia. We can directly reuse the main MICROSIM codebase in trial assessment simulations to assess trial designs. In a MICROSIM subpackage [9], we have developed a simple codebase to describe trials and trial groups, enable execution of simulated trials in the MICRO-SIM framework, and enable extraction of MICROSIM data elements for analysis. So, for example, by running simulated trials of different BP-lowering strategies across different populations

selected based on baseline cardiovascular and dementia risk, we can determine which subpopulations lead to the most efficient trials. Similarly, we can explore how trial population characteristics (e.g., size, baseline cardiovascular and dementia risk) interact with trial durations, given the non-linear risk of dementia at older ages. Third, simulation inputs are conceptually well-aligned with existing epidemiologic data sources and concepts. The aim underlying our motivating study questions is to estimate the change in an individual over time and then aggregate the individual-level results to the population level—the same basic concept motivating cohort studies. The primary downside to this approach is that it requires considerable coding complexity and computational execution time relative to simpler modeling approaches. Here, we describe our approach to developing our individual and population-level simulation model of ASCVD and dementia and our attempt to mitigate some general limitations facing simulation analyses.

## Simulation design principles

Simulation analyses are implemented by developing software codebases that encode the simulation logic and stochastic elements for data creation. These codebases are often large, containing tens of thousands of lines of computer code or more. As codebases increase in size, so does the expected number of bugs [10]. Ensuring that simulation analyses produce accurate and reliable results requires sound software development processes such that the code implements the underlying simulation logic with sufficient code quality. To this end, we developed a series of principles and strategies based on computer programming best practices to guide MICROSIM development: Transparency, Readability, Simplicity, Testing, and Validation [11]. We hope that by designing a simulation following these principles, simulation reliability and accuracy will be enhanced. We further hope that designing an extensible framework potentially adaptable to various chronic disease management problems will enable researchers to build upon this framework to maximize the reliability and valdity of subsequent analyses.

1. Transparency—the mere possibility that other groups will assess one's codebase incentivizes coders to program carefully and ensure the code is readable and coherent. The entire MICROSIM codebase [12], including the notebooks used to develop MICROSIM's inputs and the notebooks used for the validation analyses [13] presented in this manuscript, are all publicly available on GitHub [12]. Opening the repository to other investigators also broadens the user base and thereby increases the likelihood of identifying errors in the simulation and improving the rigor and reproducibility of the simulation.

2. Readability—the more readable the code, the less likely it is for errors to emerge. We used three main strategies to achieve this. First, we hired a software developer with industry experience (coauthor, LC, Ann Arbor, MI) to assist with the coding and the software development methodology. Second, the core elements of the simulation were developed via pair programming, where two coders (JB and LC) sat side-by-side and shared a keyboard to develop key simulation elements. Third, we used automatic formatting tools (Black [14]) that ran when a file was saved to have a consistent and predictable code style, making the code easier to read and write.

3. Simplicity—the simplest possible model structures and assumptions were employed whenever possible. The simpler the logic to be implemented, the less likely it would be implemented with error. This principle was intended to apply both to the simulation code and overall simulation structure. At the code level, this principle is largely a statement of priorities. If a relatively simple data structure can represent the data, then it should be preferred over a more complex data structure. Similarly, while more complex statistical models may

represent the data structure modestly better unless there are major gains in performance, simpler statistical models should be preferred as they are less likely to be implemented with error.

4. Testing—in a large codebase, it is possible that a change in one area may have unintended consequences elsewhere. We employed unit tests in a largely test-driven development paradigm to mitigate this risk [15]. That is, for a new piece of code or bug fix, first, a test was developed that would fail until the code was correctly implemented, and that same test would only pass when the code was implemented correctly. Tests also ran automatically on TravisCI [16] when new code was pushed to GitHub, making it easy to see if a change caused a test to fail.

5. Validation against the best available data—the core element to ensure accurate simulation results was to compare the simulation results to real-world data that were not included in the model derivation whenever possible. This manuscript summarizes the key validation steps that MICROSIM has undergone to date.

## Overview of simulation evidence evaluation hierarchy and structure

MICROSIM is a population and individual-based Monte Carlo simulation using varying regression-based models (e.g., logistic regression for binary outcomes, Cox regression for time-to-event data) to model annual transition probabilities in risk factors and outcomes [12]. MICROSIM is designed to enable the direct application of data from common clinical and epidemiological data sources to disentangle the complex interaction, at the individual level, between competing forces across heterogeneous individuals and understand population-level effects by aggregating across individuals.

Model assumptions and inputs were derived from the best available evidence and designed to be directly compatible and strongly conceptually linked to the best existing data sources. Specifically, MICROSIM is initialized with individuals from population-level data containing individual-level phenotypic data related to CVD. Then, transitions over time in risk factors and outcomes are modeled using regression-based data from cohort studies. Finally, trial-based evidence on treatment effects can be integrated at the population level to ensure that overall treatment effects align with trial-based evidence. The specific evidence was selected by applying an evidence hierarchy that prioritized meta-analyses of RCTs over individual RCTs over high-quality observational evidence. When none of these were available, we estimated effects using regression models, and the BP COG pooled dataset of individual participant data from six well-characterized US prospective cardiovascular cohort studies: Atherosclerosis Risk in Communities Study (ARIC) [17], Coronary Artery Risk Development in Young Adults Study (CARDIA) [18], Cardiovascular Health Study (CHS) [19], Framingham Offspring Study (FOS) [20], Multi-Ethnic Study of Atherosclerosis (MESA), and Northern Manhattan Study (NOMAS) [21]. MICROSIM is designed to model scenarios for preventing ASCVD and dementia based on prior separate simulation models estimating ASCVD [2] and dementia prevention [22] (**Fig 1**).

In MICROSIM, we examine how healthy individuals (i.e., free of ASCVD and dementia) transition into non-ASCVD death, fatal or non-fatal ASCVD, all-cause dementia, or remain free of all those events. MICROSIM updates the status of healthy individuals annually using Monte Carlo methods. The MICROSIM population is derived from the nationally representative National Health and Nutrition Examination Survey (NHANES). Risk factor levels, cognition levels, ASCVD events, non-ASCVD death, and all-cause dementia transition rates are primarily estimated using predictive models derived from the BP COG pooled cohort dataset.

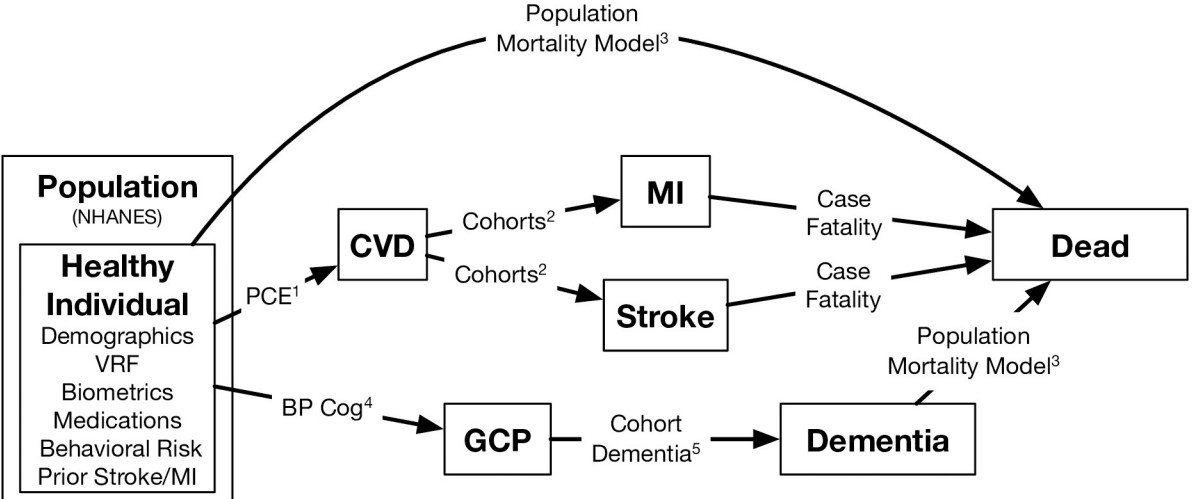

**Fig 1. Overview of MICROSIM.** Abbreviations: ASCVD, atherosclerotic cardiovascular disease. BP COG, The Effect of Lower Blood Pressure over the Life Course on Late-life Cognition in Blacks, Hispanics, and Whites (BP COG) study. GCP, global cognitive performance. MDM, the Michigan Dementia Model. MI, myocardial infarction. NHANES, National Health and Nutrition Examination Survey. PCE, Pooled Cohort Equations. VRF, vascular risk factor. [1]PCE—Effect size estimates of atherosclerotic cardiovascular disease risk is estimated with the modified Pooled Cohort Equations and used to determine if an individual has an ASCVD event in a given year. [2]Cohorts—Effect sizes were estimated using regression models and the BP COG pooled dataset of individual participant data from six US prospective cardiovascular cohort studies: Atherosclerosis Risk in Communities Study (ARIC), Coronary Artery Risk Development in Young Adults Study (CARDIA), Cardiovascular Health Study (CHS), Framingham Offspring Study (FOS), Multi-Ethnic Study of Atherosclerosis (MESA), and Northern Manhattan Study (NOMAS). Within the BP COG pooled dataset, a separate model was developed to predict MI vs. stroke for individuals with a history of ASCVD, based on all baseline characteristics. [3]Population Mortality Model—Non-CV mortality risk was estimated using a logistic regression model derived in NHANES that estimates the risk of death in a given year from non-CV causes (including dementia) and was applied separately to two groups in the population: healthy individuals with no history of dementia, stroke and myocardial infarction and individuals with dementia. [4]BP COG—Effect size estimates of the effect of risk factors on global cognitive performance were derived directly from published linear mixed-effects analyses of the BP COG study [23]. [5]Cohort Dementia—Dementia risk was estimated using a cox proportional hazards model using GCP and change in GCP as key covariates in the three BP COG cohorts that had incident dementia data (ARIC, CHS, FOS).

Treatment effects are derived directly from RCTs, as described in the "Simulation Details" sections below. Based on existing research, quality-adjusted life years (QALYs) are assigned to each state.

## Simulation details

We describe MICROSIM's simulation details using the PARTE (Properties, Actions, Rules, Time, Environment) framework developed to characterize the building blocks of agent-based models. MICROSIM should not be characterized as an agent-based model given the lack of direct interaction between individuals in the model; however, given the lack of a widely accepted standard for reporting simulation models such as MICROSIM, we felt this framework structure aligned with the key elements in MICROSIM.

## Simulation details: Properties [24]

Individuals in MICROSIM are characterized by baseline properties, largely reflecting risk factors for clinical outcomes, and by clinical outcomes that accrue over time. The baseline properties are: demographics (age, sex, race/ethnicity, education [less than high school, some high school, high school graduate, some college, college graduate]), vascular risk factors (systolic and diastolic BP [SBP, DBP], total cholesterol, triglycerides, low-density lipoprotein [LDL] cholesterol, high-density lipoprotein [HDL] cholesterol, glycosylated hemoglobin [HgbA1C]),

biometrics (body mass index [BMI], waist circumference), behaviors relevant to vascular risk (physical activity, smoking), medications (anti-hypertensive agents, statins, other lipid-lowering agents), vascular events prior to entering the simulation (stroke, myocardial infarction [MI] and cognition. Populations are characterized by the aggregation of individuals. Clinical outcomes are all-cause dementia, cognition, stroke, MI, death, and quality adjusted life years (QALYS).

To develop a population representing the overall United States (US) population and specific (e.g., condition-specific) subgroups, MICROSIM is initialized with data on adults 18 and older with no history of dementia, stroke, or myocardial infarction from NHANES. NHANES is a nationally representative serial cross-sectional survey designed to assess the health and nutritional status of the US population [25]. The survey is unique in that it combines interviews and physical examinations. NHANES collects physiological measurements of vascular risk factors using standardized protocols. Within the combined NHANES dataset, we used multiple imputations with chained equations to account for missing data. Imputation strategies for a given variable included all other variables as covariates with the following exceptions: triglycerides and LDL, which were excluded from models for each other due to collinearity, and anti-hypertensive medications which were excluded from SBP and DBP models do to collinearity. Imputation models for anti-hypertensive medications added the following: an indicator variable for whether the patients' BP qualified as hypertensive by the Eighth Joint National Committee (JNC-8) criteria, interactions between DBP, SBP, and JNC-8 hypertension status, and interactions between SBP, DBP, and self-reported hypertension to improve predictiveness.

NHANES does not repeatedly measure cognitive data, and thus cognitive model inputs were drawn from other sources. In prior work, our group built longitudinal models to predict global cognitive performance (GCP) using pooled and harmonized data from the six BP-COG cohorts (ARIC, CARDIA, CHS, FOS, MESA, NOMAS) which had repeated cognitive measures [23, 26]. In brief, trained cohort staff administered cognitive function tests longitudinally in person to participants using cognitive tests. To make inferences about cognitive domains instead of individual cognitive test items and to resolve the challenge of different cognitive tests administered across the cohorts, we co-calibrated available cognitive test items into a factor representing global cognition (global cognitive performance), using item response theory methods and confirmatory factory analyses that leverage all available cognitive information in common across cohorts and test items unique to particular cohorts [27–30]. GCP factor scores were estimated using the regression-based method in Mplus, such that a 1-point difference represents a 0.1 SD difference in the distribution of cognition across the cohorts. Higher cognitive scores indicate better performance. At baseline, each individual was assigned an overall level of GCP above/below average with a random draw from the GCP random effect distribution and initial GCP was estimated by summing the random effect with the fixed effects based on the linear predictor using baseline factors.

## Simulation details: Actions

In MICROSIM, there is no direct interaction between individuals in the current iteration of the simulation. Instead, individual-level data at prior time points are used to inform individual-level data at subsequent time points using the rules below. The exception to this rule is that treatment effects can be recalibrated at the population level after all individual-level effects have been updated as needed to align with the best available data, as described below.

## Simulation details: Rules

Most individual-level changes in MICROSIM are driven by regression models that predict the next value of a property for an individual at subsequent time points. Rules are applied

sequentially: 1. Risk factors are updated 2. Clinical outcomes are assigned based on updated risk factors 3. QALYs are assigned based on clinical outcomes. The exception is the assignment of QALYs which are assigned algorithmically.

## Simulation details: Rules—change in risk factor levels over time [31]

We built longitudinal regression models to predict levels of each time-varying individual-level factor (vascular risk factors, biometrics, vascular risk behaviors and medications) in the baseline population as the dependent variable using pooled individual participant data from the six BP-COG cohorts (ARIC, CARDIA, CHS, FOS, MESA, NOMAS). Predictor variables included race/ethnicity, smoking, gender, lagged individual-level factor level (i.e., the value of the factor at the immediately prior time point), and mean lagged individual-level factor level (i.e., mean value of the factor across all prior time points) and all other time-varying individual-level factors. Regression results were stored as JavaScript Object Notation (JSON) files, including characteristics of the residual distributions (mean and standard deviation). Advancing in one-year increments, new levels of each risk factor were calculated for each individual by summing the linear predictor (the sum of all regression coefficients * covariate values) from the corresponding regression model with a random draw from the residual distributions. SBP and DBP values were log-transformed throughout to improve model fit and de-transformed when estimating updated individual risk factor levels.

## Simulation details: Rules—cardiovascular events

For each stage of the model, an individual's risk of a ASCVD event was calculated based on their estimated vascular risk factor levels using the updated Pooled Cohort Equation (PCE), which predicts the risk of stroke, MI, or cardiovascular death [32]. To assign events, individuals, were each assigned an event with a probability based on their annual estimated risk.

After determining whether individuals would have a ASCVD event, specific event types were assigned using a separate model [33] Using individuals with a stroke or MI in the pooled BP-COG cohort, a logistic regression model was built that included factors understood to differentially predict stroke vs. MI: age, SBP, DBP, BMI, triglycerides, HgbA1C, gender, and race/ethnicity using lagged values for all time-varying individual-level factors. In the simulation, when an individual was assigned an ASCVD event, the inverse logit of the linear predictor of the stroke prediction model was used to determine the probability that the event would represent a stroke. ASCVD and events were randomly classified as stroke vs. MI using this probability. Fatal vs. non-fatal CVD event determinations were made by randomly assigning mortality for stroke (0.15) [34–37] and MI (0.13) [38].

## Simulation details: Rules—non-cardiovascular (non-CV) mortality [39]

After assigning ASCVD events, non-CV mortality was assigned using a similar approach. The risk of non-CV mortality was estimated from a logistic regression model developed using NHANES data. Specifically, long-term mortality data was linked to the combined NHANES dataset, and underlying causes of death were identified. A synthetic dataset was then developed where each NHANES respondent-year was represented by a row from the time of the initial NHANES response until the last follow-up date, and a binary indicator variable represented whether the patient died from a non-CV cause in that year. A logistic regression model was developed in this dataset predicting time to non-CV death after adjusting for age, age squared, race/ethnicity, gender, mean SBP, mean DBP, HgbA1C, total cholesterol, triglycerides, BMI, and smoking status. Then, the probability of death in a given year was estimated by taking the logit of the linear predictor from this model. Mortality was randomly assigned using this

probability; otherwise, the patient continued in the simulation alive into the next wave. When this model was included in the overall MICROSIM simulation, population-level mortality was underestimated, particularly in older adults. To account for this, we recalibrated the age and age-squared regression coefficients by running serial simulations in the overall MICROSIM environment and modifying the age and age-squared regression coefficients and measuring population-level deviation from lifetables. The coefficients that resulted in the smallest squared deviation of age-specific predicted probabilities compared to US life tables were included in the final model.

## Simulation details: Rules—cognition and dementia

We estimated each individual's future GCP values based on the individual's linear prediction from the GCP model (described above), the individual's random effect, and a random draw from the overall residual distribution.

To stratify dementia risk across risk factors, we built a Cox proportional hazards model to predict all-cause dementia using the BP-COG cohorts with data on incident dementia (ARIC, CHS, and FOS). CARDIA, MESA, and NOMAS did not have available data on incident dementia. ARIC [17], CHS [40], and FOS [41] measured incident dementia by physician adjudication using standard diagnostic criteria, study-specific protocols, and all available data, including in-person neuropsychological and neurologic assessments, telephone interviews (participant or informant), brain imaging, and medical record review. The model predicting dementia included baseline values of GCP scores, education, age, sex, race/ethnicity, and GCP slope (change in GCP scores over time) as covariates. Covariates are summarized in **Table 1**.

The incidence of dementia is somewhat higher in the combined BP-COG cohorts (ARIC, CHS, FOS) than has been observed in a prior meta-analysis of epidemiologic studies of dementia incidence and summarized in an equation by Brookmeyer et al., [42, 43] likely due to slightly different definitions of dementia. Thus, we fit a quadratic function to the baseline survival curve from the BP-COG cohort-derived Cox model and searched parameter space for modifications of the quadratic parameters such that the final dementia incidence most closely

**Table 1. Coefficients of the Cox regression model predicting time to all-cause dementia using data from six pooled US cardiovascular cohorts and incorporated in the Michigan Chronic Disease Simulation Model (MICROSIM).**

| Variable | Hazard Ratio [95% CI] |
|---|---|
| Baseline global cognitive performance score (per one-unit increase | 0.927 [0.92–0.935] |
| Slope of global cognitive performance (per one-unit increase in global cognitive score per year) | 0.999 [0.999–0.999] |
| Baseline age (per one-year increase) | 1.108 [1.101–1.114] |
| Female vs Male | 1.1 [0.995–1.215] |
| Education | |
| Eighth grade or less vs. College graduate or higher | 1.032 [0.85–1.252] |
| Some high school vs. College graduate or higher | 1.088 [0.919–1.287] |
| Completed high School/GED vs. College graduate or higher | 0.919 [0.807–1.045] |
| Some college but no degree vs. College graduate or higher | 0.797 [0.678–0.939] |
| Non-Hispanic Black vs. Non-Hispanic White | 1.214 [1.048–1.406] |

Global cognitive performance factor scores were estimated using the regression-based method in Mplus, such that a 1-point difference represents a 0.1 standard deviation difference in the distribution of cognition across the cohorts. Higher cognitive scores indicate better performance.

fits the Brookmeyer equation. This recalibration can easily be "turned off" for a given analysis if the combined BP-COG cohort incidence of dementia is thought to represent dementia incidence more accurately in the target population.

## Simulation details: Rules—treatment effects

A central goal of this simulation is to estimate how ASCVD events are impacted in the counterfactual where BP treatment differs from usual care. This goal requires reliable estimates of how an additional anti-hypertensive medication impacts risk.

MICROSIM uses effect size estimates derived directly from meta-analyses of RCTs or individual RCTs whenever available but operationalizes this by first implementing causal effects derived from observational studies and then recalibrates these effect sizes to match those observed in RCTs. In observational analyses, the association between BP levels and ASCVD events generally under-estimates the ASCVD treatment effect (i.e., relative risk [RRR] reduction) observed for anti-hypertensive medications in trials [44]. That is, an anti-hypertensive medication in a trial may lower BP by 5/3 mmHg and have a RRR for ASCVD of 0.2. However, in observational data, individuals with BPs that are 5/3 mmHg lower will have a smaller RRR for ASCVD, 0.1. Therefore, we modeled anti-hypertensive treatment effects by first applying the mean BP lowering observed across anti-hypertensive trials for each anti-hypertensive agent added—5.5/3.1 mmHg [45]. Blood pressure treatments can be applied in MICROSIM in one of two ways—they can either be applied at a single point in time or they can be reapplied during each simulation wave. In both cases, SBP and DBP in subsequent waves will reflect the lower BP in the prior wave. In this way, it is possible to model different assumptions about the life course effects of BP medications. Default behavior is to apply a treatment at a given point (e.g. add one medication to lower BP by 5.5/3.1 mmHg in a given wave) and then to have the lower BP level in that wave predict subsequent BP levels via risk factor updatingTo make ASCVD treatment effects consistent with the RCT evidence, a treatment recalibration phase at the end of each annual increment adjusted for the smaller treatment effect of an additional anti-hypertensive medication on ASCVD observed in RCTs based on the trial/observational RRR per achieved mm Hg BP reduction (relative risk, RR, of 0.79/BP medication for stroke and RR of 0.87/BP medication for MI) by randomly rolling back ASCVD events. Individuals who received an additional anti-hypertensive medication were randomly chosen to have their events rolled back, weighted by their inverse untreated risk such that the highest risk patients would be least likely to be chosen [29]. A strength of implementing treatment effects via population-level recalibration is that it enables the direct incorporation of trial effects into the codebase (e.g. post hoc modification of baseline risk by the relative risk reduction estimated in a trial) as opposed to incorporating these effects by more complex prospective mathematical approaches. The primary weakness of this approach is that it is computationally intensive and makes parallel operation of the simulation more complex as each individual can no longer be thought of as a separate Markov process.

## Simulation details: Rules—Quality Adjusted Life Years (QALYS)

QALYs were assigned to each individual in each iteration. If an individual had no events (no MI, stroke, or dementia), they were assigned an average QALY based on age [46]. For events (stroke, MI, all-cause dementia), baseline age-based utilities were reduced by relative effects for each age. For stroke and MI, the magnitude of the reduction varied such that the effect was greater in the year of an incident event (RR, 0.67 for stroke, RR, 0.88 for MI) and lower in subsequent years (RRs, 0.9 for both stroke and MI) [2]. For dementia, we used a multiplier of 0.80

for the first year of incident dementia with a 0.01 reduction in each subsequent year. (i.e., 5 years after a dementia, multiplier = 0.75) [47].

## Simulation details: Time [24]

Time is represented in MICROSIM as years. The simulation is initialized to represent a given year and mapped to an initial NHANES wave. Then, the simulation is advanced by one year and all rules are applied to each individual. After all individuals have been updated in a given wave, population-level recalibration is applied for treatment effects to ensure that treatment effects applied in the simulation reflect effects measured in randomized trials. MICROSIM can be advanced a fix number of years in time, but the default is to study the entire life course by advancing the population until all individuals have died.

## Simulation details: Environment

MICROSIM is designed to represent the US population. Thus, each simulation run is initialized with a starting year to represent the US population at a given point in time (1999–2017, including) [24] and a sample size. For a given starting year the matching NHANES wave is selected to initialize the baseline simulation properties. Initialization occurs using survey-weighted sampling with replacement from the NHANES at the specified simulation sample size. Once initialized, the baseline simulation environment does not change and does not further interact with individuals.

## Simulation details: Summary of core assumptions

The central baseline assumption of MICROSIM is that it reflects the US population at any given time point in the 21st century. Thus, the population, baseline levels of risk and unmeasured factors giving rise to both change in risk factors and outcomes over time reflect the average state of affairs in the US during this epoch. These assumptions are embedded in both the population-level data used to initialize the simulation and the data driving the underlying risk and outcome models selected for MICROSIM. A key implication of this assumption is that "usual care" reflects care that was delivered in the United States at a give time. While central, the MICROSIM framework enables this assumption to be relaxed or modified by either initializing with a different set of population parameters and/or changing the risk factor/outcome models.

Several key assumptions regarding BP treatment are built into MICROSIM. BP treatment is assumed to have narrow and specific effects, specifically: 1. Lowering SBP and DBP by a standard average effect (5.5/3.1 mm Hg) and have a resultant small effect in reducing the risk of ASCVD events via this level of BP reduction 2. Further reducing ASCVD event risk, such that the overall relative risk reduction of BP medications reflects trial meta-analyses (relative risk, RR, of 0.79/BP medication for stroke and RR of 0.87/BP medication for MI) 3. Improving GCP and reducing dementia risk mediated only through the effect of BP lowering. As such, the effect of a BP lowering medication on GCP is mediated entirely through the effect of lowering BP by 5/3 mm Hg and the effect on dementia is mediated entirely through the GCP effect.

Treatment effects operate across the life course. When a BP treatment is applied, it reduces a simulated person's BP at a given point in time. When the person is "aged" forward, the treated BP is used to predict the subsequent BP via risk factor model updating. Thus, the effect of BP treatment over time may have larger or smaller effects over time, based on the details of the risk factor updating models. This assumption can be modified within the existing MICROSIM treatment framework by specifying alternate BP treatment effects (e.g. reducing BP every wave on treatment by 5/3 mm Hg, without updating subsequent risk factors), but the default is

to apply a BP treatment at one wave and then subsequent update BP via changes in risk factor models over time [2, 17, 24, 32–47].

## Codebase overview

The simulation is structured around two core classes—Person and Population. The Person class largely serves as a data store of risk factors and outcomes with properties and methods for getting and updating the Person's state throughout the simulation. Time-varying properties are stored on Person objects as lists and outcomes are stored as dictionaries with a key representing the clinical outcome occurred and the value representing the details of the clinical outcome The Population class manages a group of Persons. Specific Population instances are responsible for loading data from a given data source, advancing the population forward in time (i.e., by one year), recalibrating population-level outcomes, and simple reporting. To help with advancing, the Population uses Models, which compute the next property (e.g., risk factor or outcome) for a Person, which it obtains from a Model Repository, a Model store that can also determine which Model to apply to a Person. Most Models are Regression Models, which take a series of regression coefficients and estimate individual-level risks for a given Person.

Each of these elements is designed to be adaptable and extensible to new populations, new parameters, and new risk models. For example, one could initialize the simulation with novel data, and the Population class could then be subclassed, and that logic added to initialized Person objects. Similarly, it is readily possible to change which Models are used to implement the changes in specific factors over time. For example, replacing the existing ASCVD risk model with a newer or different population risk model can be easily accommodated.

## Validation methods and results

### Validation strategy

We sought to identify the key elements that would be most likely to influence how well the model corresponds with high-quality data that was not used in model development for our core constructs and central research questions. For these elements, then, we identified the best validation strategy. The core elements we focused on were overall population representativeness, CV risk factor levels over time, incidence of ASCVD events, and dementia incidence.

### Validation of baseline simulated population [48]

Table 2 compares two simulated populations from MICROSIM to published NHANES standards. We found that MICROSIM's randomly selected 500,000-person simulated population nearly identically matched published demographics in the 2007–2010 survey-weighted NHANES cohort [49]. Similarly, we found excellent matching in demographics and vascular risk factor levels in our 500,000-person simulated population with hypertension (defined as SBP > 140/90 mmHg or treated with anti-hypertensive medication) to a published survey-weighted NHANES cohort of individuals with hypertension in 2013 [50].

### Validation of the simulated population's vascular risk factor levels over time [51]

Although we found the simulation framework closely reproduced the longitudinal changes in vascular risk factor levels observed in the actual BP COG cohorts (results not shown), this finding is somewhat circular since those cohorts were used to inform the simulation's models. We, therefore, assessed how well the simulation framework reproduces the longitudinal changes in vascular risk factor levels over time by comparing a longitudinal cohort from the simulation to

**Table 2. Comparison of risk factors levels between the simulation, NHANES, and the NHANES sub-population with hypertension.**

| Variable | 2007-2009b [37] | | 2013, Hypertension (SBP > 140/90 mm/Hg or any anti-hypertensive medication) [38] | |
|---|---|---|---|---|
| | NHANES | Simulation | NHANES | Simulation |
| Age, years (mean) | 45.9 | 45.9 | 60.0 | 58.4 |
| Female | 51.7% | 51.7% | 50.0% | 49.7% |
| **Race/Ethnicity** | | | | |
| White | 68.4% | 68.3% | 71.0% | 69.6% |
| Black | 11.5% | 11.4% | 14.0% | 14.0% |
| Hispanic | 13.6% | 13.6% | 10.0% | 10.3% |
| BMI, kg/m$^2$ (mean) | 28.5 | 28.5 | 31.0 | 30.9 |
| Hypertension prevalence based on the Eighth Joint National Committee Criteria | | | 80.0% | 82.5% |
| SBP, mm Hg (mean) | | | 133.4 | 132.1 |
| DBP, mm Hg (mean) | | | 71.6 | 72.2 |
| **Medications** | | | | |
| Anti-hypertension medication use | | | 41.0% | 41.4% |
| Statin medication use | | | 41.0% | 41.4% |

Abbreviations: BMI, body mass index. DBP, diastolic blood pressure. NHANES, National Health and Nutrition Examination Survey. SBP, systolic blood pressure.

a pseudo-cohort from NHANES (derived from repeated cross-sectional NHANES cohorts) as a more robust assessment of the simulation's fidelity. Specifically, we built a simulated nationally representative population of 250,000 adults 18 or older with no history of stroke, myocardial infarction or dementia using the NHANES in 1999 and advanced the simulated population for 18 years until 2017 (the most recent year for which NHANES data was available). We then removed simulated individuals that died prior to 2017 from the population. For our NHANES comparator, we included adults age 36 (baseline age 18 + 18 years of age advancement) or older and excluded adults that immigrated into the US as the simulation does not account for in or out-migration. Histograms comparing the vascular risk factor levels estimated in the simulated population(initialized in 1999 and advanced 18 years) to those observed in the pseudo-cohort population (NHANES 2017 without in-migration) are presented in **Fig 2**. The simulation generally closely reproduced both central tendencies and variances of risk factor levels, except for over-predicting DBP levels (mean 78.7, SD 9.2 vs. 71.6 mmHg, SD 10.9) and slightly under-predicting the variance in total cholesterol levels (mean 200.0 mg/dL, SD 31.0 vs. 196.8 mg/dL, SD 41.3).

## Validation of ASCVD event incidence and mortality [52]

**Table 3** summarizes the estimated overall incidence of stroke and MI in a simulated population of 250,000 individuals from 1999–2015 from MICROSIM. Overall age-standardized annual MI incidence was 234/100,000 population 95% CI [232–241] in the simulation (initialized to 1999). MICROSIM's estimated annual MI incidence is broadly comparable to the observed annual MI incidence in the Kaiser Permanente population, which ranged from 208–284 events per 100,000 from 1999–2008 [53]. Overall age-standardized stroke incidence was 153 per 100,000 [149–156] in the simulation. MICROSIM's estimated annual stroke incidence is within the range of stroke incidence reported in population-based studies over this time course, ranging from 130-400/100,000 [54–56] with hospitalization rates around 200/100,000 [57].

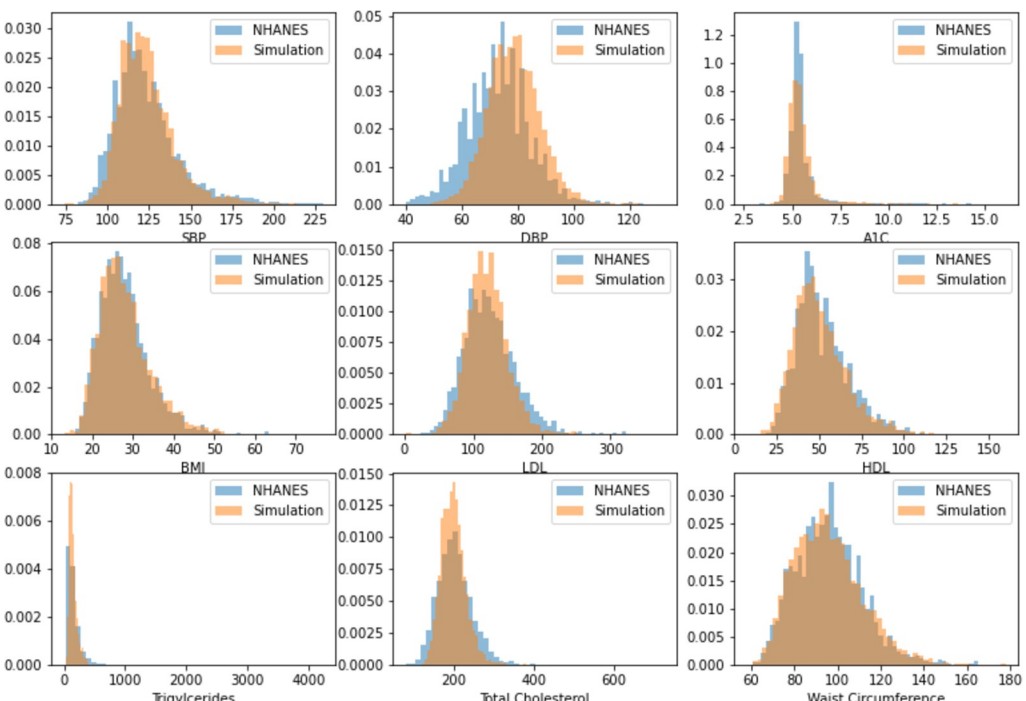

**Fig 2. Histograms of vascular risk factor level distributions in the simulation, representing the US Population, compared to a comparable NHANES sample.** Abbreviations: BMI, body mass index. DBP, diastolic blood pressure. HbA1c, glycosylated hemoglobin. HDL, high-density lipoprotein cholesterol. LDL, low-density lipoprotein cholesterol. SBP, systolic blood pressure. NHANES, National Health and Nutrition Examination Survey.

Major racial disparities in stroke incidence exist, with age-standardized stroke incidence in Black individuals generally about double the incidence in White individuals [55, 56]. In the simulation, the age-standardized incidence in Black individuals was 243/100,000 vs. 123/100,000 in White individuals, generally reproducing reported findings in the literature [55, 56]. The relatively small racial differences in MI incidence reported in the literature [58] were similarly reproduced in MICROSIM's results—incidence of 249/100,000 in White individuals vs. 219/100,000 in Black individuals.

Overall age-standardized mortality in MICROSIM is 699/100,000 compared to 729/100,000 in population-level estimates. After recalibration, MICROSIM reproduced the age-specific probability of mortality in US national life tables (Fig 3).

**Table 3. Incidence of stroke and myocardial infarction in the Michigan Chronic Disease Simulation Model (MICROSIM), overall and by race.**

| | | Epidemiologic Standard | Simulation |
|---|---|---|---|
| | | Events/100,000 Population | Events /100,000 population [95% CI] |
| Myocardial infarction | All | 208–284 | 234 [232–241] |
| | White individuals | 199 | 249 [243–254] |
| | Black individuals | 189 | 219 [204–233] |
| Stroke | All | 130–400 | 153 [149–156] |
| | White individuals | 208 | 123 [119–126] |
| | Black individuals | 331 | 243 [228–258] |

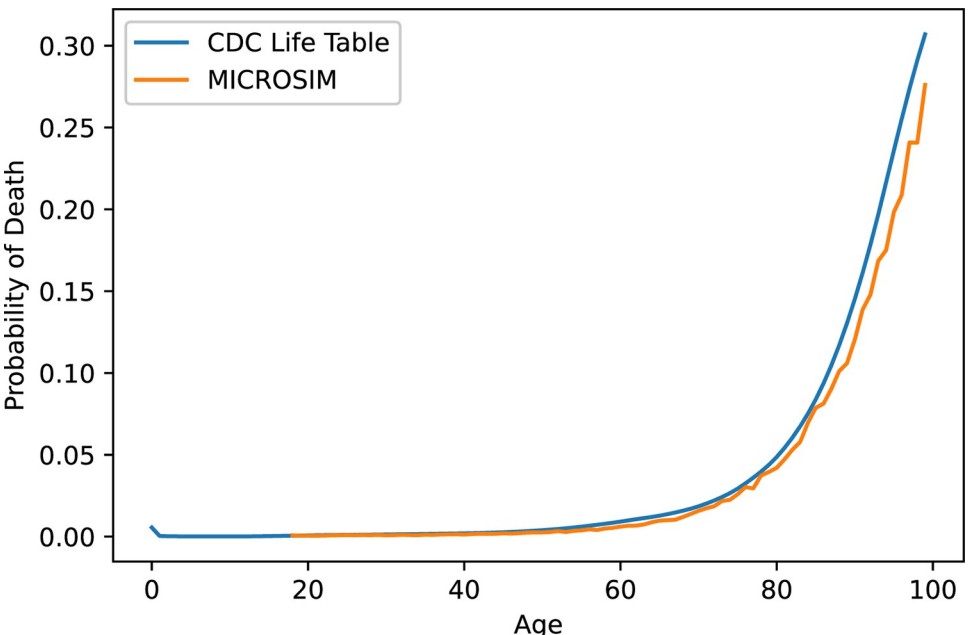

**Fig 3. Probability of death in the next year by age in the Michigan Chronic Disease Simulation Model (MICROSIM) vs. life table data.** Comparison of the probability of death by age in the simulated population vs. age-specific probability of death in Centers for Disease Control life tables [61]. Michigan Chronic Disease Simulation Model (MICROSIM).

## Validation of treatment effects [59]

To determine whether the simulation reproduced real-world BP treatment effects, we ran 15 simulations advancing a population of 150,000 individuals 18 years and older with no history of dementia, stroke, or myocardial infarction from NHANES for five years under two scenarios: an "as-treated" version of the US population where BP treatment reflects current practice (usual care) and a population where every individual added a single BP medication to their current regimen (mean BP lowering 5.5/3.1 mmHg). The simulation estimated the mean RR for stroke was 0.76 [range 0.72–0.82], and the mean RR for MI was 0.85 [range 0.81–0.89]. These estimated effect sizes compare closely to our calibration standard, derived from a meta-analysis of BP-lowering treatment trials which observed a [RR of stroke of 0.79 and a relative risk of MI of 0.87 [60].

## Validation of dementia incidence [61]

We created a nationally representative population of 200,000 individuals 18 years and older with no history of dementia, stroke, or myocardial infarction using NHANES. The simulation model advanced the population for 20 years and estimated and all-cause dementia incidence. **Fig 4** compares the estimated all-cause dementia incidence from MICROSIM to the age-dementia incidence curve from the Brookmeyer et al. meta-analysis [42]. There was a close agreement between the MICROSIM-estimated all-cause dementia incidence and the meta-analytic all-cause dementia incidence across the age spectrum.

## Illustration of the potential utility of a combined individual and population-based model framework

A potential virtue of the combined individual and population-based approach of MICROSIM is that different parameters that would typically require different study designs to estimate can

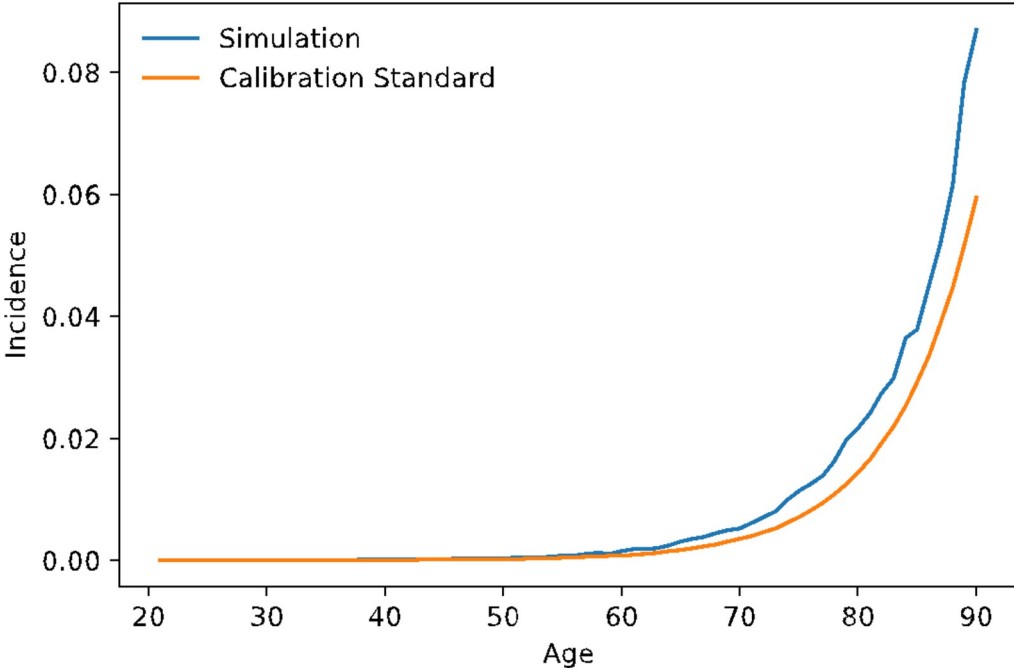

**Fig 4. Comparison of all-cause dementia incidence (incident dementia /100,000 population) between the simulation and a population standard.** Comparison of raw dementia incidence by age in the simulated population vs. summarized age-specific incidence from the Brookmeyer et al., meta-analysis [33].

be estimated within the MICROSIM framework. For example, conventional fixed probability decision analyses are designed to estimate population level mean effects. (e.g. for a given BP treatment strategy, how many QALYs are gained compared to usual care?). Similarly, regression-based analysis of cohort studies can estimate between- and within-group variance of different BP treatment strategies. MICROSIM's design allows both types of parameters and their interaction to be estimated.

To illustrate how the combined individual and population-based modeling framework used in MICROSIM can be used to characterize population-level effects, individual-level effects, and effects stratified by variables of interest, we ran a simple model comparing a 300,000-person sample of individuals 18 or older with baseline hypertension and no history of dementia, stroke, or myocardial infarction treated with usual care and another model with a similar sample of 300,000 individuals with baseline hypertension treated with intensive BP control based on the Systolic Blood Pressure Intervention Trial (SPRINT) [5]. MICROSIM estimated a small gain in population-level mean QALYS with intensive BP control compared to usual care (20.9 vs. 20.8) similar to the type of result that would obtain from a conventional fixed probability decision analysis. However, we also determined that the magnitude of individual-level mean QALY gains varied across the population (**Fig 5**). There was no difference in QALYs among individuals in the lowest quartile of baseline BP level (32.2 QALYS vs. 32.2 QALYs), but a considerably larger gain in those at the highest quartile of baseline BP (13.7 QALYs vs. 13.4 QALYs). The distributions within each of these quartiles (**Fig 6**) demonstrates that the small differences in mean QALYs between groups represent a very small fraction of the overall variance in individual-level QALYs, similar to the type of result that would be obtained by a regression-based analysis of a cohort study.

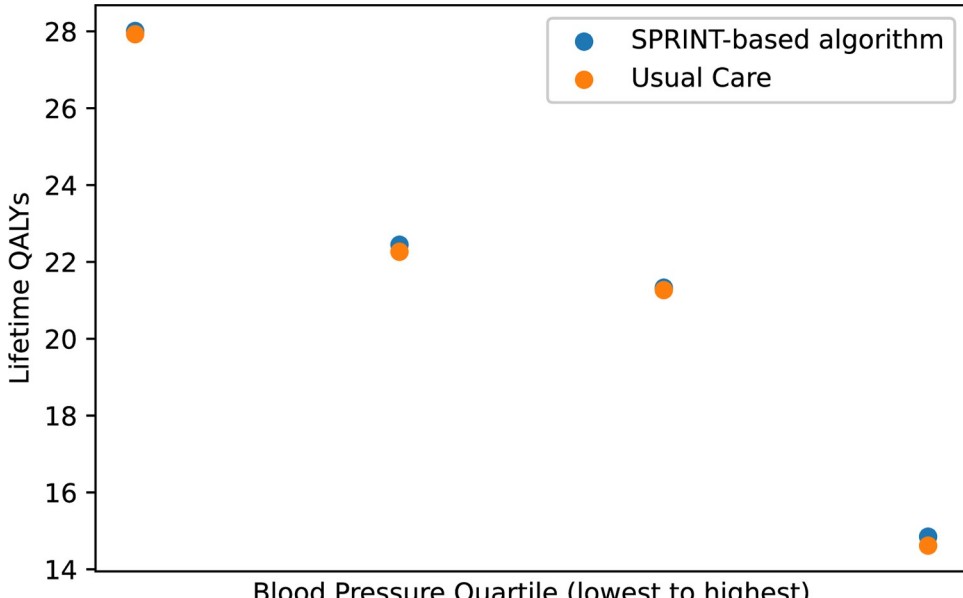

**Fig 5. Comparison of mean lifetime quality adjusted life years with population-level implementation of an intensive blood pressure control treatment algorithm compared to usual care across quartiles of baseline systolic blood pressure using the Michigan Chronic Disease Simulation Model (MICROSIM).** Abbreviations: QALY, quality adjusted life year. SPRINT, Systolic Blood Pressure Intervention Trial. Intensive blood pressure control was based on the Systolic Blood Pressure Intervention Trial (SPRINT-based Algorithm).

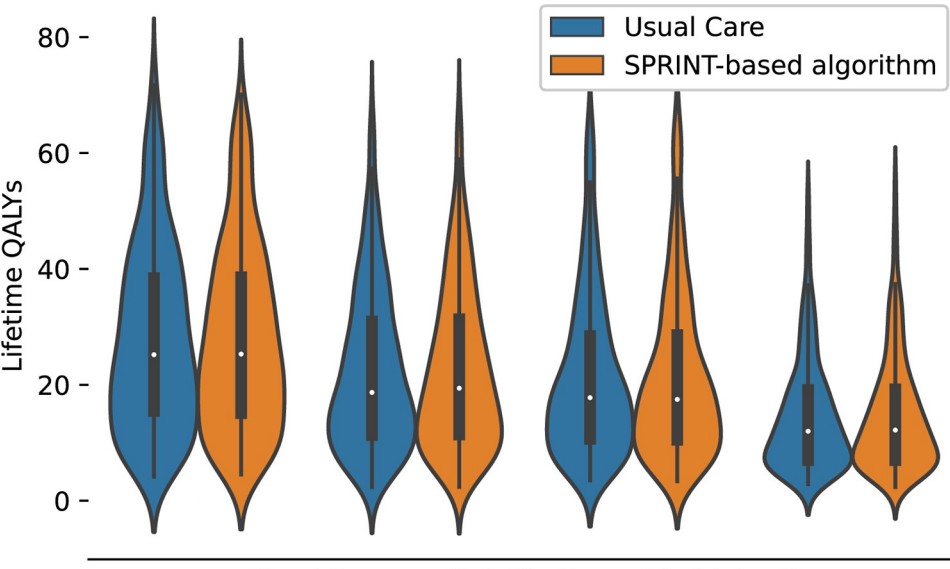

**Fig 6. Comparison of the simulation's estimated lifetime quality adjusted life years distributions with population-level implementation of an intensive blood pressure control treatment algorithm compared to usual care across quartiles of baseline systolic blood pressure using the Michigan Chronic Disease Simulation Model (MICROSIM).** Abbreviations: QALY, quality adjusted life year. SPRINT, Systolic Blood Pressure Intervention Trial. Intensive blood pressure control was based on the Systolic Blood Pressure Intervention Trial (SPRINT-based Algorithm).

## Discussion

We describe the creation of a novel microsimulation model of ASCVD and dementia. MICROSIM is an extensible, open-source population-based simulation model initially designed to explore cardiovascular and cognitive outcomes with varying approaches to BP treatment. The core elements needed for those goals validate well against population-level standards. The simulation is structured to enable relatively easy modifications (e.g., changing specific risk models) and to be extended to add additional outcomes and/or structure within clinical outcomes.

In its current iteration, MICROSIM has the elements in place to address a set of research questions around vascular risk factors, risk factor management, ASCVD, dementia, and quality of life. It is relatively easy to slightly modify this basic framework, though, to address a variety of related research questions. For example, projecting future ASCVD and dementia under different risk assumptions and/or definitions of ASCVD and dementia or assessing the cost-effectiveness of BP treatment with and without valuation of cognition. Perhaps of greater significance, the extensible framework opens the possibility of more comprehensively accounting for other disease states (e.g., congestive heart failure, peripheral vascular disease, chronic kidney disease), risk markers (e.g., social determinants of health), treatments (e.g., statins, oral hypoglycemics) and public health relevant interventions (e.g., access to nutrition, green spaces). Through such extensions, this simulation could address a vast array of research questions. In ongoing work, we are also developing frameworks to readily simulate clinical trials and add disease-specific phenotypic information. Specifically, we are expanding the simulation to assess post-stroke vascular risk factor management by including stroke type, ischemic stroke subtypes, and severity.

MICROSIM's core strengths are the application of optimal software design principles in its development, the use of the best available data to derive core simulation assumptions, and the external validation of its core elements. The application of programming best practices is, to our knowledge, relatively unique for an academic simulation, and we believe this will result in a sufficiently reliable codebase to generate accurate population-level inferences.

MICROSIM's core limitations are that, as with any simulation, the results are only as strong as the core model assumptions. Given its complexity, MICROSIM relies on many assumptions that are difficult to directly evaluate given the current state of evidence (e.g., the effect of BP treatment on cognition). As stronger evidence emerges, our intent is to continue to incorporate the best available evidence into MICROSIM. Similarly, the model structure assumes relatively simple relationships when reality may be more complex (e.g., the potential J-shaped curve relating DBP to mortality). Thus, optimal application of MICROSIM to specific research questions will require consideration of how those assumptions may influence results. When uncertain model assumptions can plausibly influence conclusions, robust sensitivity analyses could be conducted by altering those assumptions and reassessing results. Additionally, while race was included in the regression models underlying MICROSIM, the data used to derive these regression models was often insufficient to address the effects of Hispanic ethnicity (e.g., on dementia). While Hispanic individuals are included in the model, it is not clear that MICROSIM would reproduce societal-level estimates of outcomes in Hispanic individuals.

## Author Contributions

**Conceptualization:** Deborah A. Levine.

**Data curation:** Rachael Whitney.

**Formal analysis:** Rachael Whitney.

**Funding acquisition:** Deborah A. Levine.

**Methodology:** Deborah A. Levine.

**Project administration:** James F. Burke, Deborah A. Levine.

**Software:** Luciana L. Copeland.

**Supervision:** James F. Burke, Deborah A. Levine.

**Visualization:** Deborah A. Levine.

**Writing – original draft:** James F. Burke, Jeremy B. Sussman, Rodney A. Hayward, Alden L. Gross, Emily M. Briceño, Rachael Whitney, Bruno J. Giordani, Mitchell S. V. Elkind, Jennifer J. Manly, Rebecca F. Gottesman, Darrell J. Gaskin, Stephen Sidney, Kristine Yaffe, Ralph L. Sacco, Susan R. Heckbert, Timothy M. Hughes, Andrzej T. Galecki, Deborah A. Levine.

**Writing – review & editing:** James F. Burke, Jeremy B. Sussman, Rodney A. Hayward, Alden L. Gross, Deborah A. Levine.

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
