## [Decision Letter · Decision Letter 0]

28 Jul 2023

PONE-D-23-00713Development and Validation of the Michigan Chronic Disease Simulation Model (MICROSIM)PLOS ONE

Dear Dr. Burke,

Thank you for submitting your manuscript to PLOS ONE. After careful consideration, we feel that it has merit but does not fully meet PLOS ONE’s publication criteria as it currently stands. Therefore, we invite you to submit a revised version of the manuscript that addresses the points raised during the review process.

We look forward to receiving your revised manuscript.

Kind regards,

Jussi Olli Tapani Sipilä, M.D., Ph.D., B.Soc.Sci.

Academic Editor

PLOS ONE

Journal Requirements:

This research project is supported by a grant R01 NS102715 from the National Institute of Neurological Disorders and Stroke (NINDS), National Institutes of Health (NIH), Department of Health and Human Service (DHHS). The NINDS was not involved in the design and conduct of the Study; collection, management, analysis, and interpretation of the data; preparation, review, or approval of the manuscript; and decision to submit the manuscript for publication. The content is solely the responsibility of the authors and does not necessarily represent the official views of the National Institute of Neurological Disorders and Stroke or the National Institutes of Health. Additional support was provided by the National Institute of Aging (NIA) grant R01 AG051827 and 1 RF1 AG068410-01 (DL), NIA Claude Pepper Center grant P30 AG024824 (AG), NIA grants K01 AG050699 (AG), and NIA Michigan Alzheimer's Disease Research Center grant P30 AG053760 (BG).

Cohort Funding/Support: The Atherosclerosis Risk in Communities (ARIC) study has been funded in whole or in part with federal funds from the National Heart, Lung, and Blood Institute (NHLBI), NIH, DHHS, under Contract nos. (HHSN268201700001I, HHSN268201700002I, HHSN268201700003I, HHSN268201700004I, HHSN268201700005I,). Neurocognitive data are collected by U01 2U01HL096812, 2U01HL096814, 2U01HL096899, 2U01HL096902, 2U01HL096917, and R01 AG040282 from the NIH (NHLBI, NINDS, NIA, and 

National Institute on Deafness and Other

Communication Disorders). Further information about ARIC’s funding can be found at https://sites.cscc.unc.edu/aric/desc. The authors thank the staff and participants of the ARIC study for their important contributions. 

The Coronary Artery Risk Development in Young Adults Study (CARDIA) is conducted and supported by the NHLBI in collaboration with the University of Alabama at Birmingham (HHSN268201800005I & HHSN268201800007I), Northwestern University (HHSN268201800003I), University of Minnesota (HHSN268201800006I), and Kaiser Foundation Research Institute (HHSN268201800004I). A neurocognitive ancillary study is funded by the NIH (NIA, R01 AG063887). A full list of CARDIA collaborators and other cohort study information can be found at https://www.cardia.dopm.uab.edu/. This manuscript has been reviewed by CARDIA for the scientific content.

The Cardiovascular Health Study (CHS) was supported by contracts HHSN268201200036C, HHSN268200800007C, HHSN268201800001C, N01HC55222, 379 N01HC85079, N01HC85080, N01HC85081, N01HC85082, N01HC85083, N01HC85086, 75N92021D00006, and grants U01HL080295 and U01HL130114 from the NHLBI, with additional contribution from the NINDS. Additional support was provided by R01AG023629, R01AG15928, and R01AG20098 from the NIA. A full list of principal CHS investigators and institutions can be found at https://chs-nhlbi.org/CHSOverview.  The content is solely the responsibility of the authors and does not necessarily represent the official views of the National Institutes of Health.

The Framingham Heart Study is a project of the NHLBI of the NIH and Boston University School of Medicine. This project has been funded in whole or in part with federal funds from the NHLBI, NIH, DHHS, under contract No. HHSN268201500001I. 

The Northern Manhattan Stroke (NOMAS) study has been funded at least in part with federal funds from the NIH, NINDS by R01 NS29993. 

The Multi-Ethnic Study of Atherosclerosis (MESA) was supported by contracts 75N92020D00001, HHSN268201500003I, N01-HC-95159, 75N92020D00005, N01-HC-95160, 75N92020D00002, N01-HC-95161, 75N92020D00003, N01-HC-95162, 75N92020D00006, N01-HC-95163, 75N92020D00004, N01-HC-95164, 75N92020D00007, N01-HC-95165, N01-HC-95166, N01-HC-95167, N01-HC-95168, and N01-HC-95169 from the NHLBI, and by grants UL1-TR-000040, UL1-TR-001079, and UL1-TR-001420 from the National Center for Advancing Translational Sciences (NCATS). Cognitive testing at Exam 6 in MESA has been funded by grants R01HL127659 from the NHLBI and NIA R01AG054069 from the NIH. The authors thank the other investigators, the staff, and the participants of the MESA study for their valuable contributions. A full list of participating MESA investigators and institutions can be found at http://www.mesa-nhlbi.org. 

Reviewers' comments:

Reviewer's Responses to Questions

**Comments to the Author**

1. Is the manuscript technically sound, and do the data support the conclusions?

Reviewer #1: Partly

2. Has the statistical analysis been performed appropriately and rigorously? 

Reviewer #1: I Don't Know

3. Have the authors made all data underlying the findings in their manuscript fully available?

Reviewer #1: Yes

4. Is the manuscript presented in an intelligible fashion and written in standard English?

Reviewer #1: Yes

5. Review Comments to the Author

Reviewer #1: I see that this is a paper outlining the construction of a modeling framework with the intent of allowing simulation-based approaches in exploring population-level impacts of different types of interventions. The current version of the code-base allows for interventions specifically around blood pressure and other risk factors. The intended benefits are to provide a tool that can supplement more traditional approaches, like RCTs, that provide strong evidence but have large, sometimes impossible, requirements given the possible health questions.

Let me start by saying that I strongly appreciate the focus on software design best practices. Pair programming, solid testing patterns, etc are vital for building the kind of complicated software projects that could be so valuable when trying to understand the many inter-related issues in population and public health. So thank you for voicing this.

That being said, I have a few concerns.

A small first point is that the model name, MICROSIM, is almost the same as the common short-form for micro-simulation modeling. I would suggest changing the model name to avoid confusion.

A larger point, though, is that this is not what is typically understood to be an agent-based model. A key part to an ABM is some kind of interaction, either with other agents, with a shared environment, or both. And a key goal is generally to look for some kind of emergent property at the population level. As far as I can tell, this is certainly an individual-based model, but there is no inter-agent interaction, nor a shared environment. I don’t see either of these in this model.

Also, in ABMs, elements of the domain being modeled are generally labelled as endogenous, exogenous, or ignored. Endogenous elements are those whose mechanism is captured in some way in the model (e.g., an SEIR model of infection describes an infection mechanism). In MICROSIM, all dynamics are determined using cohort variables (e.g., age, sex, risk factors) and various statistical correlations derived from regression models. In other words, it sounds like each "person" is effectively a lone Markov process where the annual transition probabilities between different observable disease or risk states are determined using regression-based estimates. This could be described as an individual-based model, a micro-simulation model, or a Markov chain model, but it isn’t an agent-based model.

A good resource that describes agent-based and other models is a report called “Best Practices for Systems Science Research” on the OBSSR website, available here: https://obssr.od.nih.gov/sites/obssr/files/inline-files/best-practices-consensus-statement-FINAL-508.pdf

Now, as a Markov chain model, it is unclear to me how current methods of regression are unable to provide the same kinds of insight that are purported to be generated from this model. I appreciated the section called “Illustration of The Potential Utility of Agent-Based Model Framework” that seeks to highlight a specific example of value added by this model. But I am unable to judge the level of that contribution. Could one not already glean that population-level impacts are likely to be much lower than group-level variance by just using the regression-based hazards? What does MICROSIM add to this? I would find it helpful to see this compared with what could be done without the model to better highlight its unique contributions.

More broadly, I don’t see how this model offers more than what’s already available with regression-based statistical approaches. I would benefit from other examples on how this model has been used, or can be used, to provide useful insights, especially if they are compared to current state-of-the-art approaches. And if it was already described in the text but I missed the relevant sections, I would benefit from an indication of which sections I missed and an expansion on that context. I was very intrigued by this sentence: “We can directly reuse the MICROSIM code in the trial assessment simulations to assess trial designs.” This is an area I think could be really approachable by simulation models. But I would need to see solid examples to motivate the use of this framework.

It also strikes me that there are already other well established packages for doing agent-based modeling. (I know I already argued that this isn’t ABM, but if something can do ABM, it can also do individual-based Markov chain models.) A presentation of a new tool kit that is built to occupy a role that is arguably already being filled by, say, Repast, NetLogo, or AnyLogic should do some work to compare it against the ABM tools already available, especially since those tools also allow the creation of proper ABM internal mechanism. What does this offer that current ABM frameworks don’t?

Perhaps a better organization of this work is for the main report to better motivate the model, with strong examples of how this offers unique value over the statistical methods required to build the model, and over current ABM modeling frameworks. The technical descriptions of which parts were estimated in what specific way, of the order of simulation steps, etc could be put into a supplementary technical report or users manual. (There are ways of describing ABMs, such as the PARTE or ODD frameworks, linked in that OBSSR report, but one could also provide more traditional software design approaches like class diagrams.)

I recognize that my requests would likely require quite substantial re-writes, but I think they would be necessary.

There are some other smaller questions I had:

• What are the computational details of this package? E.g., what is the memory footprint for running the examples in the paper?

• I strongly appreciate the goal of improve transparency. How does one do that for non-coders? You had to go out to industry to find someone with software design experience. I’m not sure I see how this solves the black-box science barrier for the majority of the professional audience of this work.

• It was stated in several places that a key metric of success is how “accurate” this model is, but how accurate can it be if it’s parameterized using survey data? Is this even a useful goal for such models?

• In several places, the text describes how to trigger an event with certain probability (e.g., “by determining whether a [0, 1) random uniform number was below that partition threshold”). I think much of this text could be cut out and replaced with just “[some property] was assigned with a certain probability”. I think it would make the text cleaner.

• What is an “event-partitioning model”? The link is only the Github page with the python code.

• Shouldn’t table 2 have uncertainty bounds of some type to allow better comparison?

• Table 3 should have the empirical values included for easier comparison.

• There are a few spelling or grammar errors that can be easily fixed (e.g., the word “xcluded”)

6. PLOS authors have the option to publish the peer review history of their article (what does this mean?). If published, this will include your full peer review and any attached files.

Reviewer #1: **Yes: **L Kurt Kreuger

---

## [Author Response · Author response to Decision Letter 0]

6 Oct 2023

Response included as separate file.

---

## [Decision Letter · Decision Letter 1]

13 Dec 2023

PONE-D-23-00713R1Development and Validation of the Michigan Chronic Disease Simulation Model (MICROSIM)PLOS ONE

Dear Dr. Burke,

Thank you for submitting your manuscript to PLOS ONE. After careful consideration, we feel that it has merit but needs a minor adjustment before it can be accepted to be published in PLOS ONE. Therefore, we invite you to submit a revised version of the manuscript that addresses the points raised during the review process.

We look forward to receiving your revised manuscript.

Kind regards,

Jussi Olli Tapani Sipilä, M.D., Ph.D., B.Soc.Sci.

Academic Editor

PLOS ONE

Journal Requirements:

**Additional Editor Comments:**

Please see the reviewer's comments.

Reviewers' comments:

Reviewer's Responses to Questions

**Comments to the Author**

1. If the authors have adequately addressed your comments raised in a previous round of review and you feel that this manuscript is now acceptable for publication, you may indicate that here to bypass the “Comments to the Author” section, enter your conflict of interest statement in the “Confidential to Editor” section, and submit your "Accept" recommendation.

Reviewer #1: (No Response)

2. Is the manuscript technically sound, and do the data support the conclusions?

Reviewer #1: Yes

3. Has the statistical analysis been performed appropriately and rigorously? 

Reviewer #1: Yes

4. Have the authors made all data underlying the findings in their manuscript fully available?

Reviewer #1: Yes

5. Is the manuscript presented in an intelligible fashion and written in standard English?

Reviewer #1: Yes

6. Review Comments to the Author

Reviewer #1: The changes to the manuscript were very welcomed. I think the paper makes a strong contribution. I only have a suggestion for 1 more addition: What are the assumptions that are core to MICROSIM?

There are many necessary assumptions in the use case in this paper, but many of them are related to those of the sub-models (i.e., those statistical models within the "Model Repository"). Since this framework is intended to showcase a general approach, different sub-models can be used as evidence changes. There's no need in my eyes to changing this in the current manuscript.

However, I think there are 2 types of assumptions that seem to be related to the structure of MICROSIM. And, in my mind, these deserve special mention, even if briefly. One is that causal assumptions are needed to link some intervention(s) and some outcome(s). In this case I believe it's assumed that BP medication directly influences CVD and CVD events and nothing else. There are no side effects on any other relevant properties of the person. And it's assumed that BP medication impacts GCP and dementia indirectly through CVD and CVD events. There's no direct causal link between BP medication and GCP/dementia. Of course with greater evidence, these could be improved, but a set of "causal assumptions" that go beyond the data seems to be always needed.

And there are assumptions based on how the population is "aged" over time. With PARTE framework language, one of the "actions" is that some non-clinical properties (e.g., BMI) are updated at each model update point. How is this done? (This is the "rule" part. There's no problem to keeping these in the same paragraph.) Also, this means that someone taking BP medication will also see changes to other of their properties depending on how this update was done. Does it, for example, create the assumption that the history of BP doesn't directly impact future outcomes since it's only using the current state? Or that a reduction in BP through medication has the same future trajectory as having a naturally lower BP? Again these assumptions no doubt could be changed, but there always needs to be some kind of assumption like this.

I would find a brief paragraph about each would be useful, even in the discussion section, so that future users can be clear about the kinds of assumptions necessitated by this approach.

7. PLOS authors have the option to publish the peer review history of their article (what does this mean?). If published, this will include your full peer review and any attached files.

Reviewer #1: **Yes: **L Kurt Kreuger

---

## [Author Response · Author response to Decision Letter 1]

31 Jan 2024

Dear Dr. Sipilä and Dr. Krueger-

Thank you again for considering our manuscript. Responses to Dr. Krueger’s critiques are outlined below. We genuinely believe the manuscript has been strengthened through review and the productive feedback.

Thanks,

Jim Burke

Reviewer #1: The changes to the manuscript were very welcomed. I think the paper makes a strong contribution. I only have a suggestion for 1 more addition: What are the assumptions that are core to MICROSIM?

There are many necessary assumptions in the use case in this paper, but many of them are related to those of the sub-models (i.e., those statistical models within the "Model Repository"). Since this framework is intended to showcase a general approach, different sub-models can be used as evidence changes. There's no need in my eyes to changing this in the current manuscript.

However, I think there are 2 types of assumptions that seem to be related to the structure of MICROSIM. And, in my mind, these deserve special mention, even if briefly. One is that causal assumptions are needed to link some intervention(s) and some outcome(s). In this case I believe it's assumed that BP medication directly influences CVD and CVD events and nothing else. There are no side effects on any other relevant properties of the person. And it's assumed that BP medication impacts GCP and dementia indirectly through CVD and CVD events. There's no direct causal link between BP medication and GCP/dementia. Of course with greater evidence, these could be improved, but a set of "causal assumptions" that go beyond the data seems to be always needed.

And there are assumptions based on how the population is "aged" over time. With PARTE framework language, one of the "actions" is that some non-clinical properties (e.g., BMI) are updated at each model update point. How is this done? (This is the "rule" part. There's no problem to keeping these in the same paragraph.) Also, this means that someone taking BP medication will also see changes to other of their properties depending on how this update was done. Does it, for example, create the assumption that the history of BP doesn't directly impact future outcomes since it's only using the current state? Or that a reduction in BP through medication has the same future trajectory as having a naturally lower BP? Again these assumptions no doubt could be changed, but there always needs to be some kind of assumption like this.

I would find a brief paragraph about each would be useful, even in the discussion section, so that future users can be clear about the kinds of assumptions necessitated by this approach.

Response: 

Thank you for this helpful feedback. We have added a section at the end of the “Simulation Details” subsections that summarizes the core assumptions of MICROSIM: “Simulation Details: Summary of Core Assumptions”. 

The text of this section is reproduced in italics for convenience:

Simulation Details: Summary of Core Assumptions

The central baseline assumption of MICROSIM is that it reflects the US population at any given time point in the 21st century. Thus, the population, baseline levels of risk and unmeasured factors giving rise to both change in risk factors and outcomes over time reflect the average state of affairs in the US during this epoch. These assumptions are embedded in both the population-level data used to initialize the simulation and the data driving the underlying risk and outcome models selected for MICROSIM. A key implication of this assumption is that “usual care” reflects care that was delivered in the United States at a give time. While central, the MICROSIM framework enables this assumption to be relaxed or modified by either initializing with a different set of population parameters and/or changing the risk factor/outcome models.

Several key assumptions regarding BP treatment are built into MICROSIM. BP treatment is assumed to have narrow and specific effects, specifically: 1. Lowering SBP and DBP by a standard average effect (5.5/3.1 mm Hg) and have a resultant small effect in reducing the risk of ASCVD events via this level of BP reduction 2. Further reducing ASCVD event risk, such that the overall relative risk reduction of BP medications reflects trial meta-analyses (relative risk, RR, of 0.79/BP medication for stroke and RR of 0.87/BP medication for MI) 3. Improving GCP and reducing dementia risk mediated only through the effect of BP lowering. As such, the effect of a BP lowering medication on GCP is mediated entirely through the effect of lowering BP by 5/3 mm Hg and the effect on dementia is mediated entirely through the GCP effect.

Treatment effects operate across the life course. When a BP treatment is applied, it reduces a simulated person’s BP at a given point in time. When the person is “aged” forward, the treated BP is used to predict the subsequent BP via risk factor model updating. Thus, the effect of BP treatment over time may have larger or smaller effects over time, based on the details of the risk factor updating models. This assumption can be modified within the existing MICROSIM treatment framework by specifying alternate BP treatment effects (e.g. reducing BP every wave on treatment by 5/3 mm Hg, without updating subsequent risk factors), but the default is to apply a BP treatment at one wave and then subsequent update BP via changes in risk factor models over time.

In addition, we have the following to clarify the question of how treatment effects how individuals are “aged” over time in the subsection, “Simulation Details: Rules — Treatment Effects”:

“Blood pressure treatments can be applied in MICROSIM in one of two ways — they can either be applied at a single point in time or they can be reapplied during each simulation wave. In both cases, SBP and DBP in subsequent waves will reflect the lower BP in the prior wave. In this way, it is possible to model different assumptions about the life course effects of BP medications. Default behavior is to apply a treatment at a given point (e.g. add one medication to lower BP by 5.5/3.1 mmHg in a given wave) and then to have the lower BP level in that wave predict subsequent BP levels via risk factor updating.”

---

## [Decision Letter · Decision Letter 2]

20 Feb 2024

Development and Validation of the Michigan Chronic Disease Simulation Model (MICROSIM)

PONE-D-23-00713R2

Dear Dr. Burke,

We’re pleased to inform you that your manuscript has been judged scientifically suitable for publication and will be formally accepted for publication once it meets all outstanding technical requirements.

Kind regards,

Jussi Olli Tapani Sipilä, M.D., Ph.D., B.Soc.Sci.

Academic Editor

PLOS ONE

Additional Editor Comments (optional):

Reviewers' comments:

Reviewer's Responses to Questions

**Comments to the Author**

1. If the authors have adequately addressed your comments raised in a previous round of review and you feel that this manuscript is now acceptable for publication, you may indicate that here to bypass the “Comments to the Author” section, enter your conflict of interest statement in the “Confidential to Editor” section, and submit your "Accept" recommendation.

Reviewer #1: All comments have been addressed

2. Is the manuscript technically sound, and do the data support the conclusions?

Reviewer #1: Yes

3. Has the statistical analysis been performed appropriately and rigorously? 

Reviewer #1: Yes

4. Have the authors made all data underlying the findings in their manuscript fully available?

Reviewer #1: Yes

5. Is the manuscript presented in an intelligible fashion and written in standard English?

Reviewer #1: Yes

6. Review Comments to the Author

Reviewer #1: (No Response)

7. PLOS authors have the option to publish the peer review history of their article (what does this mean?). If published, this will include your full peer review and any attached files.

Reviewer #1: **Yes: **L Kurt Kreuger

---

## [Editor Report · Acceptance letter]

2 May 2024

PONE-D-23-00713R2 

PLOS ONE

Dear Dr. Burke, 

I'm pleased to inform you that your manuscript has been deemed suitable for publication in PLOS ONE. Congratulations! Your manuscript is now being handed over to our production team.

Kind regards, 

on behalf of

Dr. Jussi Olli Tapani Sipilä 

Academic Editor

PLOS ONE